# Increasing pulse pressure ex vivo, mimicking acute physical exercise, induces smooth muscle cell-mediated de-stiffening of murine aortic segments

Cédric H. G. Neutel [1✉], Anne-Sophie Weyns[2], Arthur Leloup[1], Sofie De Moudt[1], Pieter-Jan Guns [1] & Paul Fransen[1]

The mechanisms by which physical activity affects cardiovascular function and physiology are complex and multifactorial. In the present study, cardiac output during rest or acute physical activity was simulated in isolated aortic segments of healthy C57BL/6J wild-type mice. This was performed using the Rodent Oscillatory Tension Set-up to study Arterial Compliance (ROTSAC) by applying cyclic stretch of different amplitude, duration and frequency in well-controlled and manageable experimental conditions. Our data show that vascular smooth muscle cells (VSMCs) of the aorta have the intrinsic ability to "de-stiffen" or "relax" after periods of high cyclic stretch and to "re-stiffen" slowly thereafter upon return to normal distension pressures. Thereby, certain conditions have to be fulfilled: 1) VSMC contraction and repetitive stretching (loading/unloading cycles) are a prerequisite to induce post-exercise de-stiffening; 2) one bout of high cyclic stretch is enough to induce de- and re-stiffening. Aortic de-stiffening was highly dependent on cyclic stretch amplitude and on the manner and timing of contraction with probable involvement of focal adhesion phosphorylation/activation. Results of this study may have implications for the therapeutic potential of regular and acute physical activity and its role in the prevention and/or treatment of cardiovascular disease.

[1] Laboratory of Physiopharmacology, University of Antwerp, Campus Drie Eiken, Antwerp, Belgium. [2] Natural Products & Food Research and Analysis—Pharmaceutical Technology (NatuRA-PT), University of Antwerp, Campus Drie Eiken, Antwerp, Belgium. ✉email: Cedric.Neutel@uantwerpen.be

Our genetic predestination to high physical activity is nowadays challenged by our sedentary lifestyle. The lack of physical activity in Western societies is causing 6 to 10% of all deaths from major non-communicable diseases. Moreover, adopting a physically active lifestyle reduces cardiovascular (CV) mortality risk by more than 40%, thus approximating and even exceeding the benefits associated with antihypertensive or lipid-lowering drugs[1–3].

Cardiac output is strongly dependent on physical exercise intensity. Because large central elastic arteries buffer the stroke volume ejected during cardiac systole without imposing an excessive afterload, it is expected that central artery function changes with physical activity. Previously, we observed that mechanical activation of endothelial (ECs) and vascular smooth muscle (VSMCs) cells of mouse aorta contributes to maintain both EC basal nitric oxide (NO) release and low intracellular calcium ions ($Ca^{2+}$) in the VSMCs[4,5]. Furthermore, aortic VSMCs are pivotal in modulating aortic compliance and stiffness[6–9], as contraction of VSMCs in the aorta has been shown to cause pressure-dependent aortic stiffness hysteresis[10]. These phenomena may play a role in maintaining the high compliance of large arteries, especially at high pulse pressure (PP) as seen during physical activity. Therefore, this encouraged us to predict an important role for aortic VSMCs in the aortic mechanics during bouts of high intensity exercise[10].

In recent years the number of studies emphasizing the therapeutic potential of regular physical activity in the prevention and treatment of disease, especially cardiovascular, is booming. The mechanisms by which physical activity affects CV function and physiology, however, seem to be complex and multifactorial, involving multiple organs and organ systems such as endothelial function, oxidative stress, metabolic function, hormonal function (myokines, catecholamines), nervous system, glucose metabolism and smooth muscle function[1–3,11]. In an attempt to reduce this in vivo complexity and focus on the effects of increased cardiac output during physical activity on vascular smooth muscle function, we studied isolated aortic segments of mice in the "Rodent Oscillatory Tension Set-up to study Arterial Compliance" (ROTSAC) in which cardiac output during rest or acute physical activity could be simulated in well-controlled and manageable experimental conditions[6]. This set-up allows to simulate acute bouts of exercise ex vivo by applying cyclic stretch of different amplitude, duration and frequency to isolated aortic segments from healthy C57Bl/6J wild-type mice. As such, aortic compliance and stiffness can be investigated for different cyclic stretch protocols. We hypothesize that simulation of high intensity exercise affects aortic compliance and stiffness in mouse aortic segments studied ex vivo, and is dependent on cyclic stretch amplitude, duration and frequency.

## Results

### Smooth muscle cell contraction is necessary to induce de-stiffening (increased compliance) after a period of high pulse pressure.
Figure 1 shows a representative example of measurements performed in the presence of the $\alpha_1$ adrenoceptor agonist phenylephrine (PE) at 3 nM, an ineffective concentration, and at 2 µM PE, a maximally effective concentration. In this and all further experiments, basal NO release was inhibited with the endothelial nitric oxide synthase (eNOS) inhibitor L-NAME (300 µM) to exclude confounding effects of endothelial NO production on VSMC function. After attaining near steady-state "constriction" by PE, the PP was increased during 5 min from 40 to 90 mm Hg by increasing systolic pressure from 120 to 170 mm Hg while keeping diastolic pressure at 80 mm Hg (Fig. 1A, B). Immediately after the conditioning period of high PP, diameters

(Fig. 1C, D) at 80 and 120 mm Hg were larger than before the conditioning, the diameter change (Fig. 1E, F) was larger, compliance (Fig. 1G, H) increased and Petersons elastic modulus ($E_p$, Fig. 1I, J) was decreased (which we defined as "de-stiffening") compared to their isobaric measurement just before the high PP bout. All parameters returned to pre-conditioning values after about 20 to 30 min (which we defined as "re-stiffening").

Figure 2 summarizes the results for aortic segments, challenged with 35 nM and 2 µM PE. $E_p$ in control conditions was $301 \pm 4$ mm Hg and significantly rose by $60 \pm 19$ mm Hg ($p < 0.05$) after addition of 300 µM L-NAME. Supplementary addition of 35 nM and 2 µM PE, further increased $E_p$ to $540 \pm 73$ mm Hg ($p < 0.01$) and $1002 \pm 23$ mm Hg ($p < 0.001$), respectively. After a conditioning period of 10 min at equal (40 mm Hg, 80–120 mm Hg) or higher PP of 90 mm Hg (80–170 mm Hg), aortic segments were stretched again at 40 mm Hg PP (80–120 mm Hg) and $E_p$ was continuously determined (Fig. 2). The conditioning at 90 mm Hg PP caused a fall in $E_p$ at 80–120 mm Hg (Fig. 2A–C). Although the absolute amount of de-stiffening of $150 \pm 15$ mm Hg for 35 nM and $292 \pm 32$ mm Hg for 2 µM PE was significantly different (ANOVA $p < 0.01$), the relative amount of de-stiffening was $25.6 \pm 0.9\%$ and $33.2 \pm 3.6\%$ for 35 nM PE and 2 µM PE (respectively, $p = 0.09$) (Fig. 2D). Subsequently, $E_p$ returned for both PE concentrations to pre-conditioning values. Re-stiffening was nearly complete after 20 min (Fig. 2E) and occurred with time constants of $9.4 \pm 2.7$ min and $4.8 \pm 0.6$ min for 35 nM PE and 2 µM PE, respectively (Fig. 2F, $p = 0.15$). Because the (relative) amount of de-stiffening and the kinetics of re-stiffening were independent of the concentration of PE, all further experiments used the concentrations of 2 µM PE, which led to clear de-stiffening and re-stiffening effects.

### Static high pressure stretch does not cause de- and re-stiffening.
In the following experiments (Supplementary Fig. 1) conditioning of the segments occurred for 10 min with static or cyclic stretch of different amplitude in the presence of PE. There was an immediate, significant, de-stiffening after conditioning the segments with high cyclic stretch (i.e., 80–140 mmHg ($p = 0.03$) and 80–170 mmHg ($p = 0.0005$)), which returned to normal stiffness after a period of 30 min at 80–120 mmHg. Static stretch at 80, 120, 140 and 170 mm Hg did not result in de-stiffening.

### Aortic de-stiffening is affected by the pulse pressure amplitude.
To simulate different intensities of exercise, the PP was changed from 40 (80–120 mm Hg) to 60 (80–140 mm Hg), 90 (80–170 mm Hg) and 120 mm Hg (80–200 mm Hg) in the presence of PE (Fig. 3A). Without changing the PP, it was observed that $E_p$ slowly increased further ("time control 80–120", Fig. 3B). Conditioning for 4 min caused $E_p$ to significantly decrease ($p = 0.003$) right after the higher PP period and this decrease was PP-dependent (Fig. 3C). The time constant of return to pre-conditioning values after 30 min re-stiffening was not significantly affected by the applied PP ($p = 0.21$) (Fig. 3D). Alternatively, the maximal stiffness, obtained after full "re-stiffening", was significantly lower ($p = 0.03$) compared to pre-conditioning values (Fig. 3E). However, no differences were observed after performing multiple comparisons analysis.

### A constant pulse pressure of 40 mm Hg induces slight de-stiffening at higher pressures.
To test whether pressure itself has a similar effect on de-stiffening and the time course of re-stiffening, segments were subjected to a conditioning period of 5 min to 80–170 mm Hg and subsequently to different conditioning periods of 5 min at different pressures but with the same pulse

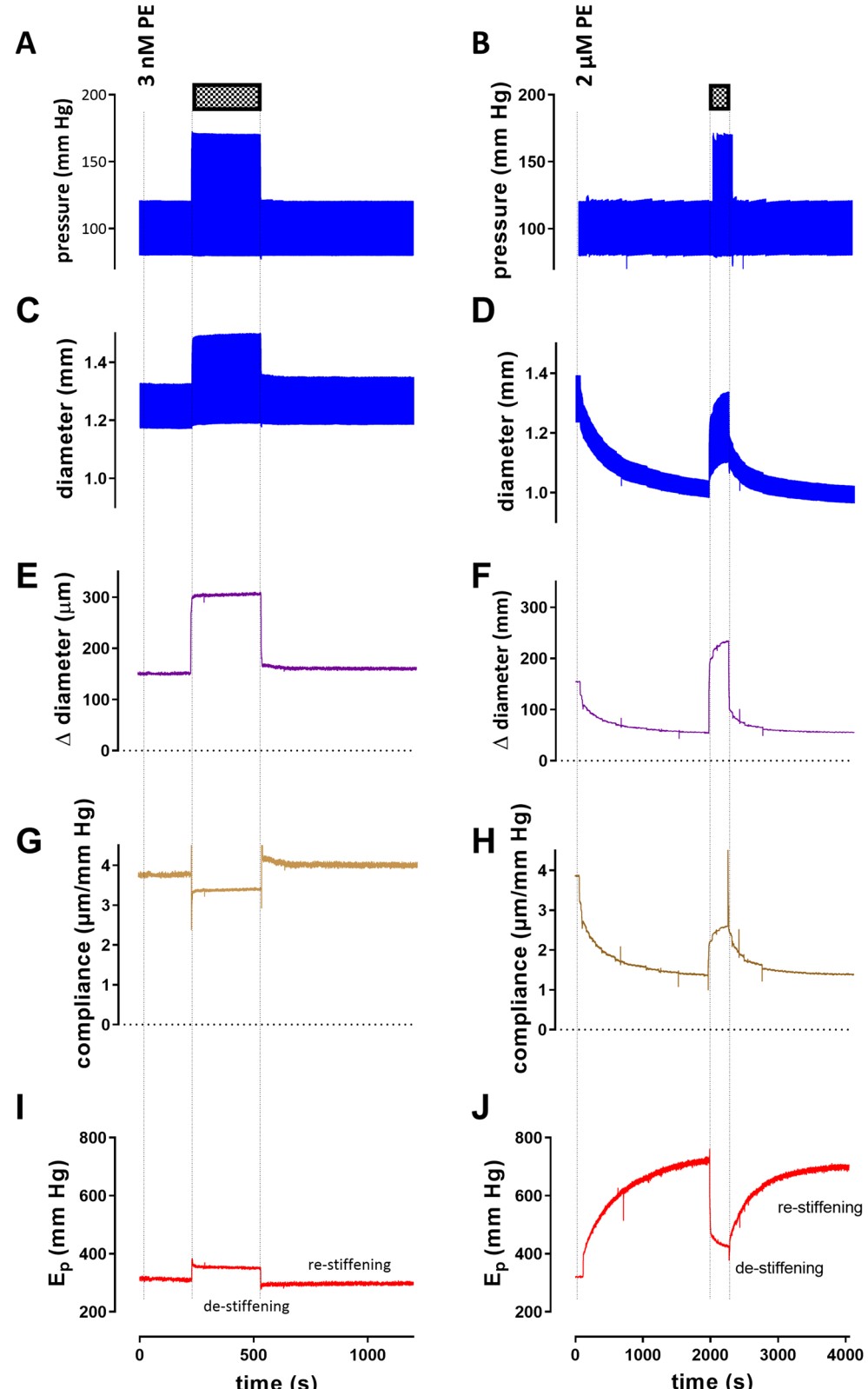

**Fig. 1 Example of an experimental protocol in which two aortic segments were stretched at a frequency of 10 Hz between 80 and 120 mm Hg in the presence 300 μM L-NAME and 3 nM PE (left) or 2 μM PE and (right).** During 5 min the pulse pressure was increased from 40 to 90 mm Hg (hatched bars), by increasing systolic pressure from 120 to 170 mm Hg and keeping diastolic pressure at 80 mm Hg (**A**, **B**). Diameters at diastolic and systolic pressures (**C**, **D**) were continuously determined. In (**E**) and (**F**), the difference between diastolic and systolic diameter is shown. When divided by the pressure difference, compliance is obtained (**G**, **H**). In (**I**) and (**J**), the Peterson modulus of elasticity ($E_p$) is displayed throughout the protocol. As indicated in (**J**), right after the conditioning period of high pulse pressure, $E_p$ was decreased (termed "de-stiffening"), followed by a return to normal stiffness (termed "re-stiffening"). PE phenylephrine; $E_p$ Peterson's modulus of elasticity.

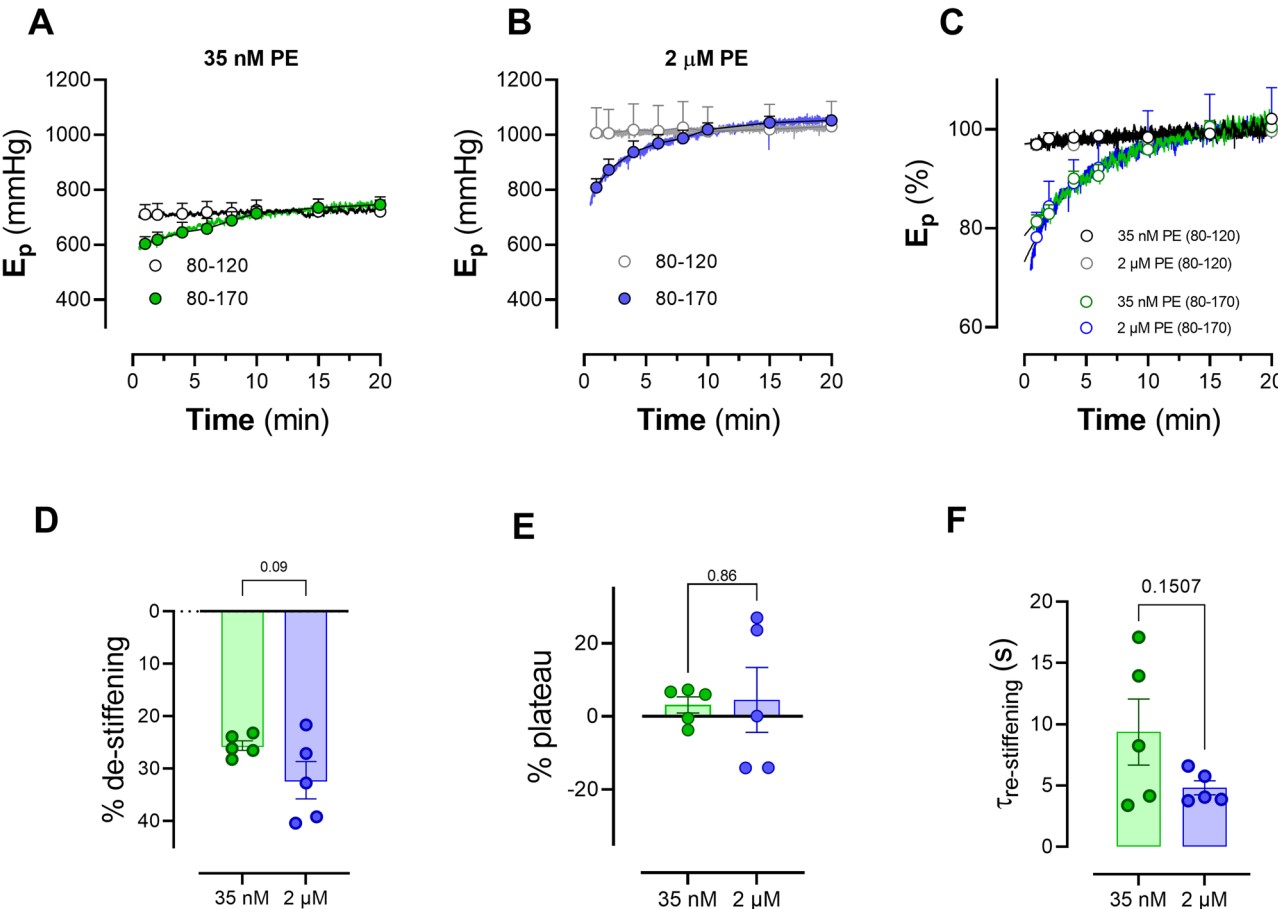

**Fig. 2 De-stiffening after conditioning at high stretch depends on the concentration of PE.** Time-dependent increase of $E_p$ of aortic segments ($n = 5$) contracted with 35 nM (green, **A**) and 2 μM PE (blue, **B**), both in the presence of 300 μM L-NAME after a conditioning period of 10 min with PP of 40 mm Hg (80–120 mm Hg, black, gray) and 90 (80–170 mm Hg, green and blue). All $E_p$ values in the figures are measured in isobaric conditions of 80–120 mm Hg after the conditioning period and were normalized to the $E_p$ values at 80–120 mm Hg in the presence of 35 nM PE (black) and 2 μM PE (gray) before the conditioning period in (**C**). The $E_p$ traces of (**A–C**) are the mean trace for 5 mice and the mean ± SEM is only indicated for certain data points. Individual traces of (**A–C**) were fitted with a mono-exponential function: $E_p = E_{p(0)} + (E_{p(30)} - E_{p(0)}) * (1 - \exp(-\text{time}/\tau))$ with $E_{p(0)}$, $E_p$ at time 0 min, $E_{p(30)}$, $E_p$ at time 30 min, and $\tau$, the time constant of re-stiffening ($\tau_{\text{re-stiffening}}$). **D** displays % de-stiffening which is the amount of de-stiffening ($100 - (E_{p(0)}/E_{p(30)}) * 100$). **E** shows the relative plateau attained after 20 min with respect to the pre-conditioning value of $E_p$. **F** displays time constants of re-stiffening. **C** Two-way ANOVA (repeated measures by both factors) with Tukey's multiple comparisons test (*, **, ***: $p < 0.05$, 0.01, 0.001 versus 80–120 mm Hg). **D–F** paired Student's $t$ test with the $p$ value indicated in the figures. PE phenylephrine, $E_p$ Peterson's modulus of elasticity.

pressure of 40 mm Hg, i.e., 80–120, 100–140, 120–160 and 140–180 mm Hg (Fig. 4). The comparison between a conditioning period of 80–170 mmHg and 100–140 mmHg is of particular interest, since both will result in the same "mean" pressure but a different pulse pressure.

In the presence of PE, $E_p$ at 80–120 mm Hg was 819 ± 43 mm Hg ($n = 5$). After conditioning for 5 min to 80–170 mm Hg, $E_p$ of the segments decreased to 562 ± 24 mm Hg (about 58% de-stiffening) and returned back to the original values with a time constant of 5.9 ± 0.6 min. Subsequently, aortic segments were conditioned for 5 min at 80–120, 100–140, 120–160 and 140–180 mmHg (box in Fig. 4B), keeping the pulse pressure constant but changing diastolic and systolic pressure. $E_p$ was again determined at 80–120 mm Hg during 30 min and expressed as relative value ($E_p$ in the presence of PE before conditioning was set to 100% in Fig. 4B). It is evident that at 80–120 mm Hg there was no change of $E_p$ (time control). At higher pressures of 100–140 and 120–160 mmHg, there was negligible de-stiffening (Fig. 4C) and, hence, the amount of re-stiffening (Fig. 4D) was small (respectively 11.1 ± 1.4 and 19.6 ± 1.8%) and not significant ($p = 0.35$). Re-stiffening also occurred with a significantly faster

time course compared to the 80–170 mmHg condition (Fig. 4E). Conditioning at very high pressures of 140–180 mm Hg caused high variability in de-stiffening, which was not significantly different from de-stiffening after 80–170 mm Hg. However, the amount of re-stiffening was significantly smaller and faster than after 80–170 mm Hg. Since there were only negligible effects after 100–140 mmHg, opposed to 80–170 mmHg, it can be stated that the contribution of mean pressure to aortic de-stiffening is limited.

**A high pulse pressure of 90 mmHg induces de-stiffening at all mean pressures, but is more pronounced at higher pressure.** To test whether a constant PP of 90 mm Hg had similar effects when varying diastolic and systolic pressure, PE-treated segments were conditioned for 4 min at 40–130, 60–150, 80–170 and 100–190 mm Hg (protocol, Supplementary Fig. 2a). Re-stiffening (with $E_p$ before conditioning set to 100%) followed a mono-exponential time course (Time × pressure interaction effect $p = 0.0464$) (Supplementary Fig. 2b) with de-stiffening (Supplementary Fig. 2c) increasing with mean pressure (above 60 mm Hg diastolic pressure) and the time constant of re-stiffening

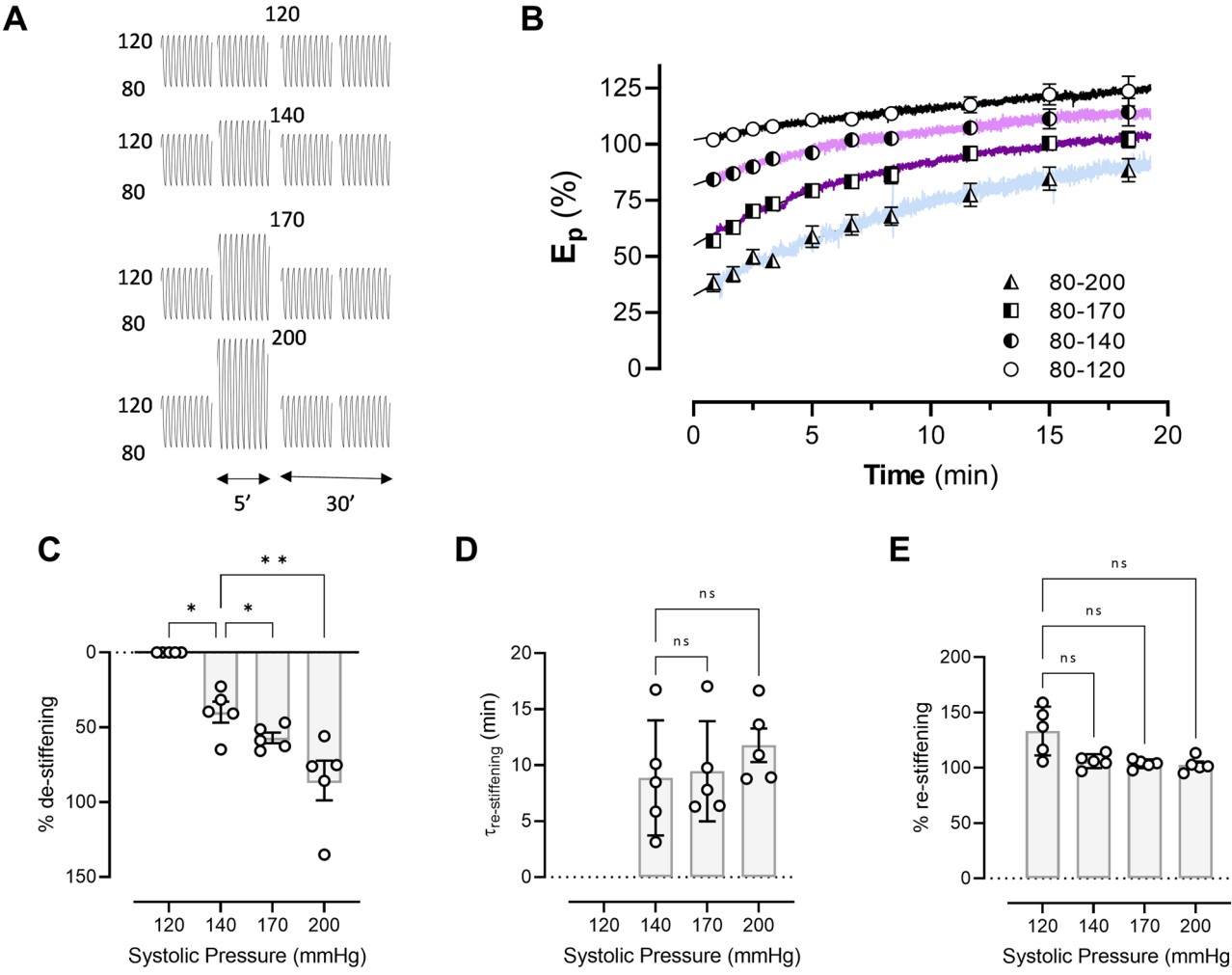

**Fig. 3 Aortic de-stiffening is dependent on pulse pressure.** Segments were contracted with 2 μM PE in the presence of L-NAME and subjected to conditioning at PP of 40, 60, 90 and 120 mm Hg as shown in the experimental protocol (**A**). The relative $E_p$-values (with respect to pre-conditioning $E_p$ values) at PP of 40 mm Hg (80–120 mm Hg) as a function of time (**B**). Traces are mean traces for five mice with certain data points as mean ± SEM (symbols). A mono-exponential fit of the individual traces of (**B**) revealed % de-stiffening (relative amount of decrease with respect to pre-conditioning values) (**C**), time constant of re-stiffening (**D**) and final amount of stiffness (relative amount of the plateau values of the exponential fit, with respect to the pre-conditioning values (**E**). $n = 5$. One-way ANOVA with post hoc test for (**C**–**E**): *, **, ns: $p < 0.05$, 0.01 and non-significant. $E_p$ Peterson's modulus of elasticity.

displaying a tendency to increase at 100–190 mm Hg (Supplementary Fig. 2d). Hence, higher mean pressure caused larger de-stiffening after conditioning at 90 mm Hg PP. Similar to our previous results, where high pressure itself only has minor effects on de-stiffening, unless it is combined with a high pulse pressure as well, as is shown here.

**Duration of the conditioning period affects de-stiffening.** Next, we investigated if the duration of high pulse pressure (to simulate exercise duration) affected subsequent de-stiffening of the aortic segments. In these experiments (Supplementary Fig. 3a), $E_p$ was measured at 80–120 mm Hg in control conditions, after addition of PE and following a conditioning period at 80–170 mm Hg with different duration (2, 5, 10 and 15 min). Indeed, the duration of the pulsatile bout significantly (ANOVA $p < 0.0001$) affected de-stiffening of aortic tissue. Remarkably, the de-stiffening amplitudes did not significantly differ with increasing duration of the conditioning at 80–170 mm Hg ($-255 \pm 33$, $-254 \pm 38$, $-292 \pm 31$ and $-326 \pm 71$ mm Hg for 2, 5, 10 and 15 min conditioning periods) (Supplementary Fig. 3c). After the conditioning period, the return to the stiffness values before conditioning

was complete for the 2 and 5 min conditioning (respectively $27 \pm 13$ and $-7 \pm 12$ mm Hg), but was incomplete for 10 and 15 min pre-conditioning (respectively $-53 \pm 37$ and $-114 \pm 56$ mm Hg) after 30 min of restoration time. These results indicate that after conditioning with durations longer than 5 min, the return to normal stiffness takes longer than 30 min. The time constants of the mono-exponential re-stiffening after the conditioning period ($7.3 \pm 1.4$, $7.6 \pm 0.9$, $7.3 \pm 1.3$ and $6.8 \pm 1.3$ min for 2, 5, 10 and 15 min conditioning) were not significantly different. In summary, whereas the duration of the high pulsatile bout does not affect the amplitude of de-stiffening, it does alter the re-stiffening to normal baseline values.

**Frequency of stretch has an impact on de-stiffening, but only at extremely low frequencies.** Because viscous properties of blood vessels are dependent on the frequency of vessel stretch, different frequencies of aortic segment stretch were tested. Besides the "normal" mouse heart rate of 10 Hz, we also investigated conditioning at 20, 1 and 0.1 Hz (Supplementary Fig. 3b). In the presence of PE, de-stiffening at 1, 10 and 20 Hz was respectively $-317 \pm 36$, $-205 \pm 24$ and $-291 \pm 46$ mm Hg and was not

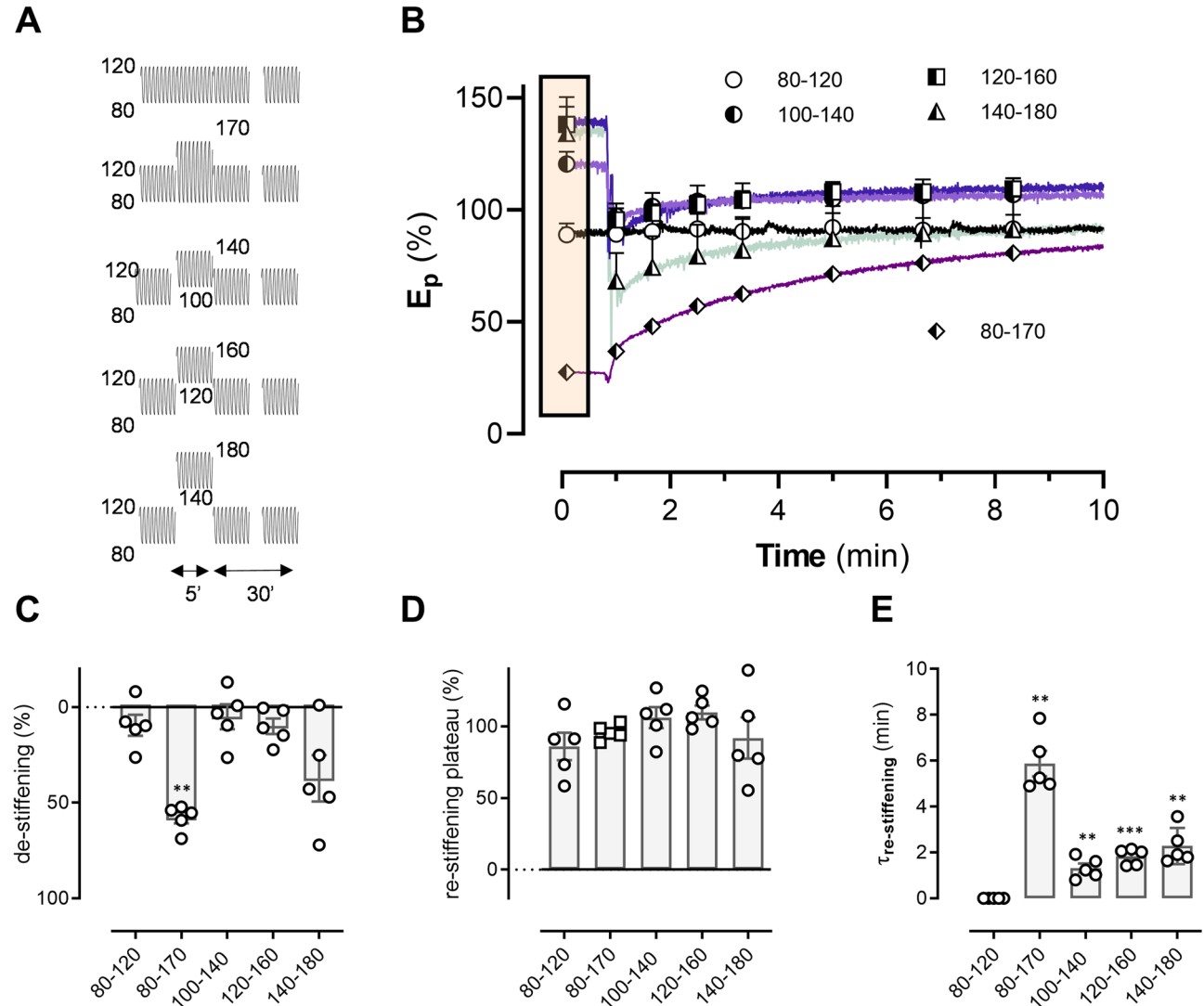

**Fig. 4 The role of mean pressure in aortic tissue de-stiffening.** Relative $E_p$ in the presence of 300 μM L-NAME and 2 μM PE as a function of time at 80–120 mm Hg after conditioning the segments for 5 min at different pressures. Experimental protocol showing that the pulse pressure was kept constant (at 40 mm Hg) whilst the mean pressure was increased: 80–120, 100–140, 120–160 and 140–180 mm Hg. Data were compared with the higher pulse pressure of 90 mm Hg (80–170 mm Hg) (**A**). $E_p$ was expressed in % with $E_p$ before the conditioning period as 100% (**B**). Curves were fitted with a mono-exponential function revealing amplitude of de-stiffening (at 50 s in the graph, when segments were clamped between 80 and 120 mm Hg) (**C**), amount of re-stiffening (**D**) and time constants of re-stiffening (**E**). The box on plot (**B**) represents the conditioning period for 5 min at 80–120 mm Hg, 100–140 mm Hg, 120–160 mm Hg and 140–180 mm Hg. *, **, ***: $p < 0.05$, 0.01, 0.001 versus 80–170 mm Hg ($n = 5$). $E_p$ Peterson's modulus of elasticity.

significantly different between the conditions (Supplementary Fig. 3d). De-stiffening after conditioning at 0.1 Hz was −179 ± 45 mm Hg and significantly smaller than at 1 Hz. All $E_p$ values recovered to near normal stiffness after 30 min. The time constants of this recovery were not significantly different and were 5.2 ± 0.7, 7.1 ± 0.8, 7.1 ± 0.7 and 6.0 ± 1.0 min after conditioning at 20, 10, 1, 0.1 Hz respectively.

**The number of bouts of high cyclic stretch has only minor effects on de- and re-stiffening.** To investigate whether de- and re-stiffening processes were dependent on the number of conditioning periods (use-dependent effect), one or multiple bouts of 5 min high cyclic stretch between 80 and 170 mm Hg were separated by periods of 5 min of normal cyclic stretch between 80 and 120 mm Hg (recovery period) (Supplementary Fig. 4a). Every bout of "exercise" caused de- and re-stiffening of the PE-treated

aortic segments (Supplementary Fig. 4b). The recovery period of 5 min, which is roughly the time constant of re-stiffening, was too short to allow complete re-stiffening, but did not result in greater de-stiffening upon the next bout (Supplementary Fig. 4b). Re-stiffening after the first, second, third and fourth bout of high cyclic stretch is superposed in Supplementary Fig. 4c and demonstrates that $E_p$ was significantly decreased after the final bout, but that there were no significant differences between the four conditions (Supplementary Fig. 4d). Also the time constant of re-stiffening was not significantly affected (Supplementary Fig. 4e). Re-stiffening was only complete after one bout (Supplementary Fig. 4f), but was incomplete 30 min after conditioning with 2, 3 and 4 bouts.

**Receptor- versus non-receptor-mediated contractions.** The above experiments showed that contraction before conditioning is

necessary to observe aortic de-stiffening by high amplitude stretch. To test whether non-receptor contractions also result in aortic de-stiffening induced by high stretch conditioning, the non-receptor depolarization with 50 mM $K^+$ (to activate voltage-gated calcium channels (VGCC)) was compared with 100 nM PE as a reference contractile agent.

In Fig. 5, the increase of arterial stiffness is compared between PE and high $K^+$. $E_p$ increased from $368 \pm 12$ in control to $383 \pm 15$ mm Hg following addition of L-NAME and to 763 mm Hg 15 min after addition of 100 nM PE. For 50 mM $K^+$, $E_p$ increased from $354 \pm 8$ mm Hg to $372 \pm 9$ mm Hg ($p < 0.001$) in the presence of L-NAME and further to $536 \pm 27$ mm Hg ($p < 0.001$) 15 min after application of 50 mM $K^+$. Hence, the effect of PE on aortic stiffening was about twice the effect of depolarization. Conditioning the segments for 10 min at high pulse pressure (80–170 mm Hg) caused a de-stiffening upon return to normal pressure of 80–120 mm Hg of $-263 \pm 25$ mm Hg for PE and $-75 \pm 6$ mm Hg for high $K^+$ (Fig. 5A, B). Indeed, there was a significant ($p < 0.0001$) difference in pulse pressure-mediated de-stiffening between PE contracted and $K^+$ contracted aortic tissue. During the subsequent period of stretching at 80–120 mm Hg, $E_p$ slowly returned to normal stiffness values in case of $\alpha_1$ adrenoceptor stimulation and depolarization (Fig. 5C). Re-stiffening occurred significantly slower following contraction by depolarization than following contraction by PE. Therefore, receptor-mediated VSMC contraction results in a greater de-stiffening after high amplitude stretch opposed to non-receptor depolarization-induced contraction.

**Re-stiffening is reduced and delayed when contraction is induced during the high PP bout.** Supplementary Fig. 5a compares two conditions in which PE was applied before or during the conditioning period of 4 min at 80–170 mm Hg (named "before" and "during" in the figure). When PE was applied before the conditioning, $E_p$ increased from $363 \pm 20$ mm Hg to $768 \pm 39$ mm Hg. Application of the high PP decreased $E_p$ to $468 \pm 12$ mm Hg. After the conditioning, $E_p$ displayed re-stiffening as shown in Supplementary Fig. 5b. In the other condition (i.e., the "during" condition), $E_p$ was $346 \pm 10$ mm Hg and increased to $387 \pm 8$ mm Hg during the high PP conditioning. Indeed, there was a significant ($p < 0.0001$) difference in de- and

re-stiffening phenomena when PE was added before or during the pulsatile bout. Addition of PE at 80–170 mm Hg did not further increase $E_p$ ($396 \pm 16$ mm Hg) and after the conditioning $E_p$ slowly increased to $581 \pm 21$ mm Hg ($p < 0.05$ versus before, Supplementary Fig. 5b). Although the aortic segments in Supplementary Fig. 5b experience the same experimental conditions, de-stiffening (Supplementary Fig. 5c), time constant of re-stiffening, but not the amount of re-stiffening (Supplementary Fig. 5d, e) and final plateau after re-stiffening (Supplementary Fig. 5f) are significantly different emphasizing the importance of the timing of PE application. This suggests that addition of PE at low PP (before) or at high PP (during) caused different effects on the biomechanics of aortic segments upon return to physiological 80–120 mm Hg conditions. When PE is applied to the aortic segments during the high PP period, re-stiffening is only about 50% of normal re-stiffening (Supplementary Fig. 5b) and occurs significantly slower (Supplementary Fig. 5d).

**Src inhibition with PP2 suggests a role of focal adhesion in arterial de-stiffening by high PP.** Focal adhesion (FA) between VSMCs and their extracellular matrix has been described to play an important role in contraction and, probably, also in arterial stiffening. FA turn-over is not static, but dynamic and is regulated by the Src family of tyrosin kinases. Their activation leads to phosphorylation of several FA proteins and downstream extra-cellular signal-regulated kinase (ERK) activation. In general, vasoconstrictor stimulation leads to SRC-dependent FA activation, which determines the rate and amplitude of $\alpha_1$ adrenoceptor agonist-induced tone development in vascular tissue. Src can be inhibited by the small molecule inhibitor PP2, which prevents PE-induced FA protein phosphorylation[12].

In the present study the effects of 10 µM PP2 on arterial stiffness, the de-stiffening effects of high PP and the re-stiffening upon return to normal PP was investigated for contractions induced by depolarization (50 mM $K^+$) and by $\alpha_1$-adrenoceptor stimulation with 100 nM PE. PP2 caused a small decrease of stiffness in baseline conditions (Fig. 6A, B, KR).

Segments were contracted before incubation of the segments with 10 µM PP2. Both high $K^+$ and PE increased segment stiffness (see Fig. 6A, B). Incubation of the segments for 20 min with PP2 resulted in a significantly ($p < 0.05$) lower stiffness with

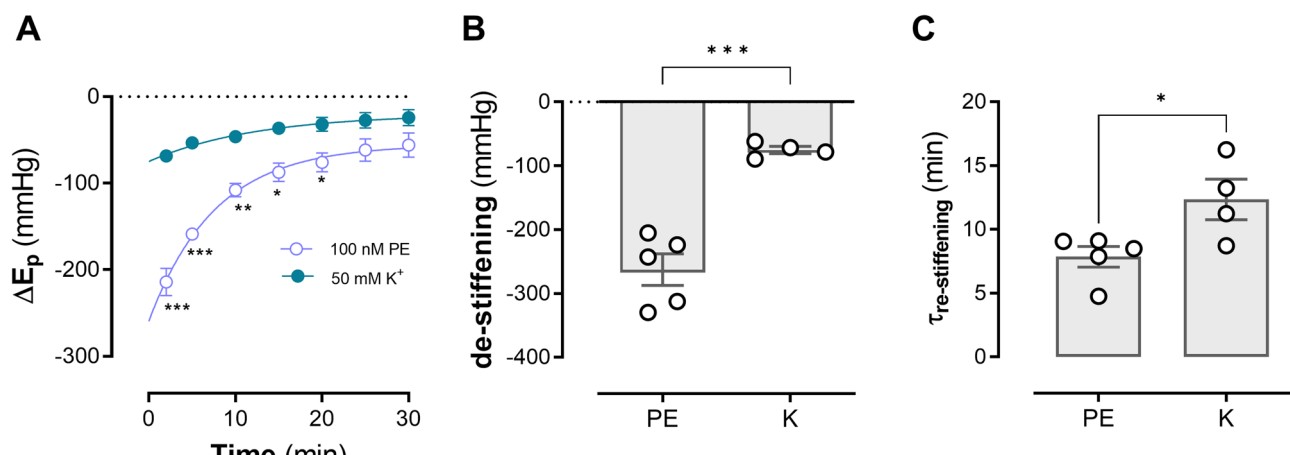

**Fig. 5 $E_p$ depends on the type of VSMC contraction.** $E_p$ was measured at 80–120 mm Hg in the absence and presence of 50 mM $K^+$ (green) and 100 nM PE (white). After conditioning at 80–170 mm Hg for 10 min, stretch was re-set to 80–120 mm Hg and the change of $E_p$ ($\Delta E_p$) was measured as a function of time in the different conditions (**A**). In (**B**), the absolute amount of de-stiffening at 80–120 mm Hg is expressed, whereas in (**C**) the time constant of re-stiffening for PE and $K^+$ is shown. Two-way ANOVA with Tukey's multiple comparison test for (**A**), unpaired $t$-test for (**B, C**). $n = 5$. *,**,***: $p < 0.05$, 0.01, 0.001, PE phenylephrine, K potassium(chloride), $E_p$ Peterson's modulus of elasticity.

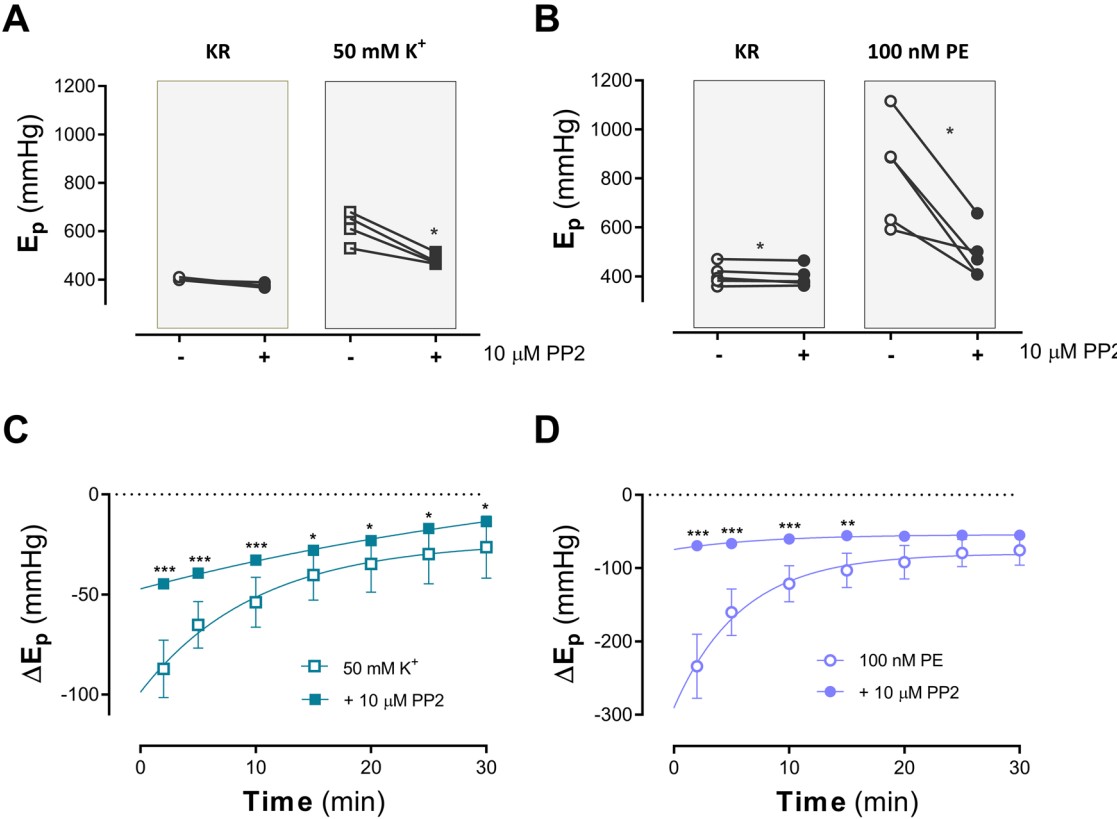

**Fig. 6 PP2 affects de- and re-stiffening after conditioning aortic segments at high pulse pressure.** The effects of 10 μM PP2 on $E_p$ were determined at 80–120 mm Hg in the absence (Krebs Ringer; KR) and presence of 50 mM K+ (**A**, $n = 4$) or 100 nM PE (**B**, $n = 5$). **C** (K+) and **D** (PE) show the time-dependent change of $E_p$ ($\Delta E_p$) as a function of time in the different conditions after conditioning the segments at 80–170 mm Hg for 10 min. Two-way ANOVA with Sidak's multiple comparison test *, **, ***: $p < 0.05$, 0.01, 0.001. PE phenylephrine, K+ potassium(chloride), $E_p$ Peterson's modulus of elasticity.

both contractile agents. Conditioning the segments for 10 min with high PP stretch caused significant ($p < 0.0001$) de-stiffening at return back to 80–120 mm Hg in all conditions, but the de-stiffening was attenuated by PP2 (Fig. 6C, D). Re-stiffening was almost completely inhibited for the PE/PP2 combination, suggesting that Src inhibition with PP2 not only abolished activation of FAs during contraction (stiffening), but also during re-stiffening.

**High PP inhibits FAK phosphorylation, which is reversed after return to normal PP.** PE increases focal adhesion activity as seen by a significant ($p < 0.05$) increase in phosphorylated Focal Adhesion Kinase (pFAK—Tyr397) (Fig. 7). Moreover, 1 min after the high pulsatile bout, a significant ($p < 0.05$) decrease in pFAK was observed compared to the contracted segments which were not subjected to this bout. The amount of pFAK normalized after 5 and 15 min.

## Discussion
The mechanisms by which physical activity affects cardiovascular function and physiology are very complex and multifactorial. In an attempt to reduce this in vivo complexity, we studied isolated aortic segments of mice in the ROTSAC in which cardiac output during rest or acute physical activity could be simulated in well-controlled and manageable experimental conditions[6]. Our data show that VSMCs of the aorta have the intrinsic ability to "de-stiffen" or "relax" after periods of high cyclic stretch and to "re-stiffen" slowly thereafter upon return to normal distension pressures. However, certain conditions have to be fulfilled: (1)

VSMC contraction is necessary to induce decreased stiffness (increased compliance) after a short period of high PP; (2) cyclic stretching (loading/unloading cycles) is a prerequisite to induce post-exercise de-stiffening; (3) de-stiffening depends on the PP amplitude, whereas the duration and stretch frequency of the conditioning period or stretch frequency have only minor impact on de-stiffening; (4) one bout of high PP is enough to induce de- and re-stiffening and the number of bouts of high cyclic stretch has only minor importance and (5) arterial de-stiffening by acute increase of vessel stretch depends on the way and timing of contraction with probable involvement of FA phosphorylation/activation.

Sex differences were not systematically looked for in the current study. While there might be differences between both sexes in the magnitude of the responses, our data in males and females never differed in the way the aortic segment responded to the different experimental protocols.

Viscoelasticity is an inherent property of smooth muscle cells (SMCs) in different tissues (gastrointestinal system, respiration system, circulation) and is a combination of pure elasticity and viscous resistance to stretch. In the circulatory system, this resistance to volume increase or stretch by ventricular ejection dampens the pulsatile pressure generated by the heart. Dampening is more important in the central, elastic arteries than in muscular arteries. By having considerable compliance and low stiffness, elastic arteries have a physiological function in protecting the downstream smaller blood vessels in delicate organs from high PP and stretch. It is evident that during physical activity cardiac output increases substantially (>4 times). Systolic blood pressure increases, diastolic blood pressure hardly changes

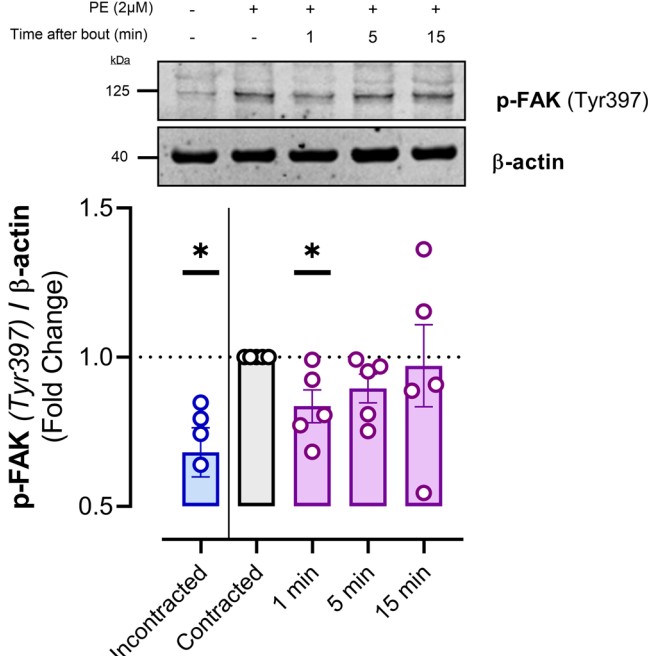

**Fig. 7 High pulsatility acutely and reversibly reduces FAK phosphorylation.** The phosphorylation state of whole aorta focal adhesion kinase/FAK(at Tyr397) was investigated before and after VSMC contraction and during 15 min after acute high pulsatile stretch (4 min at 80–170 mmHg diastolic and systolic pressure respectively). VSMC contraction induces pFAK whereas 1 min after acute high pulsatile stretch the amount of pFAK is decreased. pFAK increases again after 5 and 15 min, indicating the reversibility of stretch-induced FAK dephosphorylation. Samples derive from parallel experiments and gels/blots were performed in parallel as well. Statistics was performed using a one sample $T$-test on Log2 transformed values (hypothetical value = 0), $*p < 0.05$. $n = 5$. (p)-FAK (phosphorylated)-focal adhesion kinase.

and PP is strongly elevated. Moreover, in the healthy heart, sympathetic release of norepinephrine (NE) during physical activity increases cardiac output (>5 times higher circulating catecholamine concentrations). Although peripheral resistance in the exercising muscles decreases by relaxation of the SMC, the high PP originating from the left ventricle must be dampened as much as possible, the afterload of the left ventricle kept as small as possible and the compliance of the elastic arteries as large as possible to protect the heart and the downstream blood vessels against this temporary high PP. Elastic blood vessels may do this in different ways.

One way is by producing a high amount of relaxing factor. We have previously shown that elastic, unlike muscular arteries, produce large amounts of relaxing NO in non-stimulated conditions and that cyclic stretch promotes basal NO release[4,13,14]. Addition of PE at concentrations equivalent to the circulating catecholamine levels in vivo, hardly affect compliance or stiffness of aortic segments with "normal" endothelial function and even high concentrations of PE only moderately increase stiffness at pressures up to 150 mm Hg[9]. Therefore, it can be assumed that basal NO release in central arteries protects against circulating catecholamine induced stiffening of the aorta during exercise. Importantly, basal NO release was inhibited throughout all the experiments of this study and therefore does not affect our experimental results.

Another way of protecting the heart and downstream vessels during high cardiac output is by the viscous properties of the

VSMC leading to central artery de-stiffening after periods of high PP. VSMC in the arterial wall contribute to arterial compliance or stiffness by two dynamic components. On the one hand, there is the attachment of cycling cross bridges in the contractile filaments. On the other hand, there is the regulated transmission of force and stiffness through a non-muscle actin cytoskeleton connected to FA complexes. These FAs are integrin-containing, multi-protein structures that are part of a powerful intracellular machinery orchestrating mechanotransduction pathways, i.e., transducing environmental mechanical signals to biochemical cascades[15]. FA signaling pathways participate in regulating arterial stiffness. In the aorta, the activity of Src family kinases and Crk-associated substrate (Cas), proteins which are involved in FA dynamics, regulate FA function and cytoskeleton organization and, consequently, modulate contractility[8,12,16]. The non-muscle actin skeleton and FAs have both been shown to display plasticity in the presence of agonists and biomechanical stimuli by exhibiting stimulus-induced increase in actin polymerization and endosomal-dependent remodeling of a subset of FA proteins[17]. Both components are known to be calcium-dependent, but whether mechanical stimuli such as cyclic stretch contribute to the dynamic regulation of FAs and actin polymerization is unclear. At least, FAs are not static structures but re-organize in response to contractile agents with prominent changes in the actin regulatory proteins such as zyxin and α-actinin. This dynamic re-organization occurs on a time scale of seconds to minutes: the lifetime of proteins within the FA complex is in the order of seconds, while FAs as a subcellular structure remain stable for several tens of minutes[18]. There seems to be an important role for stretch in FA dynamics. Stretch of epithelial airway cells causes dramatic disassembly of FA and results in the redistribution of paxillin (a focal adhesion complex molecule) to the peri-nuclear region[19]. In a human bone osteosarcoma epithelial cell line, cell stretch caused two responses at different timescales: rapid FA growth within seconds after stretching, and delayed FA disassembly and loss of cell polarity that occurred over tens of minutes[20]. Tonic stretch of alveolar Type II cells may transiently disassemble FAs to unload mechanical forces in order to escape stress failure[21]. Therefore, in the present study, we hypothesized that high PP, i.e., high stretch, induces FA de-activation, whereas return to normal PP after the conditioning period with high PP may allow FA activation. Interestingly, the de-stiffening effect of high PP periods is contraction-dependent, stretch amplitude-dependent and hardly frequency-dependent. It is attenuated by PP2, a small molecule inhibitor of Src family kinases[20] and accompanied by FAK dephosphorylation, suggesting that FA are involved in the viscous properties of the aortic segments. Regardless of the conditioning protocol causing de-stiffening, re-stiffening to normal stiffness values thereafter occurred with similar time constants, suggesting that the process of re-stiffening is comparable between the different experimental conditions as would be expected for a FA re-activation process.

Such SMC de-stiffening after high stretch is also observed in the respiratory system, where deep inspiration (DI) in healthy individuals produces a transient reversal of bronchoconstriction (i.e., bronchodilation). The underlying mechanism, by which DI produces bronchodilation in healthy individuals is not completely understood, but likely involves stretch-induced relaxation of airway smooth muscle (ASMC). In vivo, a deeper depth of DI produces greater bronchodilation, which has typically been explained by increased ASMC stretch[22]. Several theoretical models have been developed to explain the bronchodilator effects of DI's. In these models integrin-mediated adhesions between ASMC and their extracellular matrix (ECM) regulate how contractile forces generated within the cell are transmitted to its external environment. The formation, size, and survival of

cell-matrix adhesions (among which FAs) adapt to environmental challenges and, dependent on the adhesion state, cell stretch may induce a permanent switch to a lower adhesion states or allow a return of the system to the high adhesion state dependent on the frequency of oscillations, cytoskeletal or ECM stiffness, and FA status. These phenomena have been mentioned to explain (in part) the transient bronchodilatory effect of a DI observed in asthmatics compared to a more sustained effect in normal subjects[23].

In our experiments, $\alpha_1$ adrenoceptor stimulation with PE increased $E_p$ significantly more, caused larger de-stiffening after the high PP period and faster re-stiffening during the post-high PP period than depolarization with high $K^+$. In isometric conditions, both contractile agents caused comparable maximal isometric force increase in isolated mouse aortic segments ($\pm 15$ mN when pre-loaded at 20 mN)[4,24]. These results indicate that contraction in isometric conditions (static stretch) is different from contraction in cyclic stretch conditions (aortic stiffening)[5]. Static or cyclic stretch transduction systems are different in the aorta. Static stretch, but not cyclic stretch (physiological 10%), activates focal adhesion kinase with the involvement of Src and integrins[25]. The mechanism of contraction is indeed different for PE and depolarization/high $K^{+26}$, which may determine the way by which the segments stiffen at 80–120 mm Hg, de-stiffen after a period of high PP and re-stiffen again after return to normal cyclic stretch at 80–120 mm Hg. It is well-known that the $Ca^{2+}$ signal, induced by the contractile agents, is the primary determinant of the contraction of VSMCs, but, subsequently, the regulation of the myosin light chain phosphatase (MLCP) activity is considered to be the most important mechanism underlying the regulation of $Ca^{2+}$ sensitivity. Thereby, the $Ca^{2+}$ signal has been shown to cross-talk with the mechanisms regulating the $Ca^{2+}$ sensitivity and MLCP activity with important roles for Rho kinase, protein kinase C, CPI- 17, and MYPT1[27–30]. This $Ca^{2+}$ sensitizing process is much more important for $\alpha_1$-adrenoceptor stimulation than for depolarization with high $K^+$ and, hence, may be more important in vessel stiffening than vessel contraction. It has also been shown that Rho-linked signaling mechanisms are involved in mechanotransduction and lead to FA activation independent of a global increase in VSMC $Ca^{2+31}$. Summarized, the way of $Ca^{2+}$ influx, $Ca^{2+}$ sensitizing and $Ca^{2+}$ removal determine how contractile agents affect arterial stiffness via interactions with the cytoskeleton-focal adhesion/integrin-extracellular matrix complex.

What can be the physiological importance of the intrinsic property of aortic VSMC to de-stiffen after a period of high PP? In in vivo studies, immediately after acute exercise and unlike the decrease of flow-mediated dilation (FMD), arterial blood pressure decreases below resting values, characterizing a state of post-exercise hypotension (PEH). A post-exercise re-setting of the baroreflex, which has a similar time course as the PEH[32], a centrally-mediated decrease in sympathetic nerve activity and local vasodilator mechanisms are believed to be at the basis of this complex hypotensive response[33–37]. PEH has some properties resembling the re-stiffening processes observed in our study. The hypotension after exercise returns to normotension with a similar slow time-course, one bout of exercise is enough to cause hypotension[38] and the amplitude is dependent on the degree of exercise (amplitude of stretch). Furthermore, PEH can be detected in normotensive individuals, but it was found to be much less consistent and of lesser magnitude than in hypertensive individuals. This may be due to other compensatory mechanisms, such as the baroreflex, that are activated in normotensive subjects, and prevent the degree of PEH from affecting orthostatic tolerance. PEH occurs independent of exercise intensity[37,39]. Because hypertension is associated with endothelial dysfunction and our results indicate that de-stiffening is more evident in the absence of basal NO release, these data may offer an alternative or additional explanation. To study the mechanisms of de- and re-stiffening, which were most evident in fully contracted segments, basal release of NO was blocked by 300 µM L-NAME. In baseline conditions inhibition of NO release with L-NAME caused a small, but significant and consistent increase of arterial stiffness and $E_p$ increased by $\pm 10$ mm Hg. Although this seems to be only a minimal increase of arterial stiffness in baseline conditions, it means that low (compatible with physiologic catecholamine concentrations) concentrations of PE (10–35 nM) have an enormous impact on aortic stiffness in case of endothelial dysfunction.

How is PP-induced de- and re-stiffening related to the well-known and well-described pressure-induced myogenic tone observed in resistance arteries and arterioles? As an inherent property of resistance arteries, the myogenic response to intra-luminal pressure elevation involves a contractile response of the SMCs to keep blood flow constant and protect delicate organs from vascular insufficiencies and excessive blood flow[40]. The big difference between re-stiffening in the aorta and the myogenic response in resistance arteries, is that the former only occurred in constricted segments, whereas the latter occurs in baseline conditions and involves stretch-induced contraction. This indicates that mechanosensitive intracellular pathways may be very different in elastic and in muscular arteries. A single bout of aerobic exercise attenuated finger, brachial and aortic PP in healthy young men, which for 20 min correlated with individual changes in aortic pulse wave velocity, but not with peak calf vascular dilator capacity[41].

What is the difference between elevated PP induced by exercise or by essential hypertension? A chronic increase of PP as in essential hypertension, is considered as a hallmark of arterial aging in humans and is also characterized by a larger increase in systolic than in diastolic blood pressure. In this situation, age-dependent changes in extracellular matrix composition[42], central artery endothelial cell function and smooth muscle stiffness are accompanied with a dramatic increase in pulse wave velocity (PWV), the best currently available measurement of arterial stiffness in vivo[43]. As such, these phenomena as observed with aging or essential hypertension have a completely different base than the phenomena observed with exercise and it will be interesting to investigate how ex vivo simulation of acute exercise in exercise-trained or sedentary aged animals affects arterial stiffness.

In summary, the present study shows that ex vivo simulation of acute exercise can be performed in isolated aortic mouse segments with the ROTSAC (increased (pulse) pressure, pulse frequency and involvement of circulating catecholamines). A concept of aortic de- and re-stiffening is described occurring after periods of high cyclic stretch. The de-stiffening depends on SMC contraction and cyclic stretch (PP) amplitude. Following return to normal cyclic stretch amplitude the aortic segments slowly "re-stiffen" to pre-exercise values. Arterial de- and re-stiffening by acute in- and decrease of vessel stretch coincides FA phosphorylation/activation.

## Methods

**Laboratory animals and tissue collection**. All animals ($n = 58$ C57Bl/6J mice, 32 female, 26 male, age between 4 and 7 months) were housed in the University of Antwerp animal facility in standard cages with 12h–12h light-dark cycles, with free access to regular chow and tap water. This study was approved by the Ethical Committee of the University of Antwerp and all experiments were performed conform to the Guide for the Care and Use of Laboratory Animals, published by the US National

Institutes of Health (NIH Publication No. 85–23, revised 1996). Aortic segments were prepared and incubated in the ROTSAC equipment as described before[6]. In short, mice were euthanized by perforating the diaphragm while under deep anesthesia (sodium pentobarbital (Sanofi, Belgium), 250 mg kg$^{-1}$, i.p.). The thoracic aorta was removed and stripped of adherent tissue. The straight part of the descending aorta was cut into 4 segments of 2 mm length and immersed in Krebs Ringer (KR) solution (37 °C, 95% $O_2$/5% $CO_2$, pH 7.4) containing (in mM): NaCl 118, KCl 4.7, $CaCl_2$ 2.5, $KH_2PO_4$ 1.2, $MgSO_4$ 1.2, $NaHCO_3$ 25, CaEDTA 0.025 and glucose 11.1. The segments were mounted between two parallel wire hooks in 10 ml organ baths.

**Ex vivo aortic biomechanics (ROTSAC).** Force and displacement of the upper hook were measured with a force-length transducer connected to a data acquisition system (Powerlab 8/30 and LabChart 8, ADInstruments). Force and displacement were acquired at 0.4 kHz. Force was measured directly by the transducer. The diameter of the vessel segment ($D$) at a given preload was derived from the displacement of the upper hook ($d$), being directly proportional to the inner circumference, as shown as in Eq. (1):

$$D = \frac{2d}{\pi} \tag{1}$$

Before each experiment, $d$ and length were determined at six different preloads (10, 20, 30, 40, 50 and 60 mN) using a camera and calibrated image software. To correct for the decrease in segment length ($l$) with increased extrapolated diameter, the average length of the segment at each cycle (100 ms) was derived from the $D$-$l$ relationship using basic linear regression.

To estimate the transmural pressure that would exist in the equilibrated vessel segment with the given distension force and dimensions, the Laplace relationship was used, as shown as in Eq. (2):

$$P = \frac{F}{l.D} \tag{2}$$

with $F$ being the force, $l$ the length (~2 mm) and $D$ the diameter of the vessel segment. The preload was adjusted until the desired diastolic and systolic pressure. At all pressures, a stretch amplitude corresponding to 40 mm Hg was chosen to allow calculation of compliance and Peterson modulus. Compliance ($C$) was calculated as shown as in Eq. (3):

$$C = \frac{\triangle D}{\triangle P} \tag{3}$$

with $\triangle D$ being the difference between systolic and diastolic diameter and $\triangle P$ being the pressure difference. The Peterson modulus of elasticity ($E_p$) is a frequently used, vessel size-independent measure of arterial stiffness and was calculated as shown as in Eq. (4):

$$Ep = D_0 \cdot \frac{\triangle P}{\triangle D} \tag{4}$$

with $D_0$ being the diastolic diameter. During all experiments, the segments were continuously stretched directly after mounting them in the organ bath with a frequency of 10 Hz (interstimulus interval of 100 ms, 50 ms increase and 50 ms decrease to systolic or diastolic pressures respectively) to mimic the physiological heart rate in mice (600 bpm) and physiological pressure (~80–120 mmHg). VSMCs could be stimulated by adding the $\alpha_1$-adrenergic agonist phenylephrine (PE) (Sigma-Aldrich, Belgium) or by depolarization with 50 mM $K^+$ Krebs solution (isosmotic replacement of $Na^+$ by $K^+$). In all experiments, the relaxing effect of endothelial derived nitric oxide (NO) was inhibited by adding 300 μM NΩ-nitro-l-arginine methyl ester (L-NAME) (Sigma-Aldrich, Belgium).

Systolic or diastolic PP could be increased by increasing systolic or diastolic preload (force and, hence, systolic or diastolic diameter). Frequency of stretching could be de- or increased by lengthening or shortening the interstimulus interval up to 10,000 ms (0.1 Hz) or 50 ms (20 Hz). The clamp to systolic pressure was always 50 ms (except for the 20 Hz frequency, where diastole and systole duration were decreased to 25 ms). $E_p$ was continuously determined at all pressure steps but values were only compared for stretching between 80 and 120 mm Hg ($\Delta P = 40$ mm Hg).

**Western blotting.** Aortic segments were mounted in the ROTSAC, as previously described, and treated with or without 2 μM PE to induce VSMC contraction. Thirty minutes after the administration of PE, the contracted segments were subjected to a high pulsatile bout by increasing the systolic pressure from 120 mmHg to 170 mmHg (with a constant diastolic pressure) for 4 min. After 4 min, the pulsatility was decreased to normal conditions (120 mmHg systolic pressure). One, 5 and 15 min after the pulsatile bout, aortic segments were removed from the ROTSAC and immediately collected in laemmli buffer supplemented with β-mercaptoethanol (5%) to induce cell lysis. Samples were loaded on Bolt 4–12% bis-tris gels (Invitrogen) for gel electrophoresis, followed by wet transfer on polyvinylidene fluoride membranes. Membranes were blocked in Odyssey® Blocking Buffer (Li-Cor Bioscience) and probed with primary antibody (overnight, 4 °C). Following primary antibodies and dilutions were used: 1:1000 rabbit anti-FAK (phosphorylated Tyr397) (Abcam, ab81298), 1:5000 mouse anti-β-actin (Abcam, ab8226). Subsequentially, (IR)-conjugated secondary antibodies (anti-rabbit: IgG926-32211 and anti-mouse: IgG926-68070; Li-Cor Biosciences) were used for infrared (IR) fluorescence detection using an Odyssey SA infrared imaging system (Li-Cor Biosciences). The western blot panel pictured in this paper was derived from one single blot. Every blot represents aortic segments, subjected to the experimental protocol, that were derived from the same animal.

**Statistics and reproducibility.** All results are expressed as the mean ± SEM with $n$ representing the number of mice, taking into account that each mouse delivered four aortic segments which could be attributed to different experimental conditions in each experiment. Analyses were performed using GraphPad Prism 8.0 (GraphPad Software, La Jolla, CA, USA). The statistical tests used in the experiments are mentioned in the corresponding figure legends. A 5% level of significance was selected. Overall ANOVA $p$ values are mentioned in the results section in the text. All attempts at replication were successful.

**Reporting summary.** Further information on research design is available in the Nature Portfolio Reporting Summary linked to this article.

## Data availability
The uncropped/unedited western blot images are included in Supplementary Fig. 6. The numerical source data that make up the graphs and charts are available as Supplementary Data. Other datasets generated during and/or analyzed during the current study are available from the corresponding author on reasonable request.

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

## Acknowledgements

This work was funded by the Fund for Scientific Research (FWO)-Flanders. C.H.G.N. is a predoctoral fellow of the FWO (grant number: 1S24720N).

## Author contributions

P.F. was responsible for the conception and design of the work. P.F., C.H.G.N., A.S.W., S.D.M., and A.L. collected the data. C.H.G.N., A.S.W., S.D.M., and P.F. analyzed and interpreted the data. C.H.G.N. drafted the article under the supervision of P.F. Critical revision of the article was the responsibility of P.F. and P.J.G. All authors approved the final version of the article.

## Competing interests

The authors declare no competing interests.
