## [Peer Review File · Communications Biology]

Reviewers' comments:

Reviewer #1 (Remarks to the Author):

The work appears to be well structured, but I have some concerns:

- Beginning of the Results section: please provide details of the mouse model used, including sample size, biological/technical replicates, etc. It is unclear throughout the text what the experimental effort was.
- Results related to Figure 1: is there any statistically significant difference that should be commented on? Please check throughout the main text, as many statements are not supported by p values, leading to questions about the biological relevance of the results.
- Any difference between males and females?

Reviewer #2 (Remarks to the Author):

The authors of this manuscript conducted a study using isolated aortic segments obtained from mice in the ROTSAC system. The study aimed to investigate the effects of acute physical activity on the aortic segments. To simulate acute physical activity, the researchers utilized $\alpha 1$ adrenoceptor stimulation. Furthermore, they evaluated the response of the aortic segments by measuring the diameter, compliance, and E_p parameters.

The paper does provide new information regarding the effect of physical activity on the diameter, compliance, and E_p parameters; however, more care is needed in the reporting of some parts of it. Please see below for my concerns:

1- In general abstract and introduction parts are not written well and it is hard to understand. The authors should re-write the second half part of the Abstract, and the first page of the Introduction.

2- In the last sentence of the abstract, the authors should mention the importance of their work in the treatment of which kind of disease.

3- Why authors didn't use any control group for their study?

In the scenario they described, a control group treated with an isotonic solution instead of the $\alpha 1$ adrenoceptor agonist, phenylephrine, would have been a suitable way to evaluate their effect on the parameters and see if there is any difference between the control group and the treated group with PE.

4- In Figure 1 authors mentioned that " Immediately after the conditioning period of high PP, diameters (figure 1c, d) at 80 and 120 mm Hg were larger than before the conditioning, the diameter change (figure 1e, f) was larger, compliance (figure 1g, h) increased and Petersons elastic modulus (E_p , figure 1i, j) was decreased (which we defined as "de-stiffening") compared to their isobaric measurement just before the high PP bout. All parameters returned to pre-conditioning values after about 20 to 30 minutes (which we defined as "re-stiffening")".

Can the authors elaborate on the duration for which the diameter, compliance, and E_p parameters remained constant at their highest levels after high PP and before re-stiffening? and how was the pattern of returning to preconditioning values?

6- in figure 6, why did the authors change the PE dose to 100nM?

Reviewer #3 (Remarks to the Author):

Major claims of the paper

High-intensity exercise simulated via cyclic stretch of appropriate amplitude, duration, and frequency affects aortic compliance and stiffness in mouse aortic segments.

VSMCs of the aorta can de-stiffen after periods of high cyclic stretch and re-stiffen slowly after returning to normal distension pressures. This is conditional on VSMC contraction and repetitive stretching (loading/unloading cycles).

The de-stiffening depends on cyclic stretch amplitude, but the duration and stretch frequency of the conditioning period or stretch frequency have only a minor impact on de-stiffening.

Arterial de-stiffening by an acute increase of vessel stretch depends on the way and timing of contraction with probable involvement of focal adhesion phosphorylation/activation.

The statistical analysis used seems appropriate, as well as the graphical visualization.

As a few comments that can improve the manuscript:

The western blot on figure 7, it is not clear how the experiment was performed to obtain the fold-change. Also, the "contracted" observation points are always at 1, does this mean that the other represents the relative fold-change from the same tissue as such sample?

Figure 6, is there evidence of successful inhibition of Src? Phosphotyrosine inhibitors occasionally fail to inhibit their target, it is recommended always test the proper inhibition of the molecule to avoid making the wrong conclusions.

Figure 4 A, the letter A needs a better description.

Overall a good contribution but it requires some details to attend before publishing.

Point-by-point response to reviewers

We thank the reviewers for their constructive comments and thorough reading of our manuscript. Below we will try to answer to each comment and hope it will contribute to improvement of our submitted text.

Reviewer #1

- 1) Beginning of the Results section: please provide details of the mouse model used, including sample size, biological/technical replicates, etc. It is unclear throughout the text what the experimental effort was.

In the beginning of our Methods section, we mentioned the use of fifty-eight C57Bl/6J mice of both sexes. C57Bl/6J is a mouse model commonly used in vascular studies [1, 2]. Of each mouse, always four aortic segments were obtained which were mounted in 4 ROTSAC set-ups and which could be subjected to different experimental protocols. For example, in figure 2, showing the effect of two phenylephrine (PE) concentrations (35 nM and 1 μ M) on Ep at 80-120 and 80-170 mm Hg, we used four aortic segments each obtained from five different mice. Two segments received 35 nM PE, while conditioned at regular (80-120 mm Hg) and higher (80-170 mmHg) pulse pressure. A similar protocol was followed for the other 2 segments of the same mouse aorta, but now in the presence of 1 μ M PE.

We have mentioned now in the Methods section/Statistical analysis:

Line 568-570: “All results are expressed as the mean \pm SEM with n representing the number of mice, taking into account that each mouse delivered 4 aortic segments which could be attributed to different experimental conditions in each experiment. Analyses were performed using Prism 8.0”.

We included the sample size in each figure (hence, n representing the number of different mice) or supplementary figure legend now, except for figure 1, which is an example of a test protocol. In the original manuscript, some figure legends lacked a direct notion of the sample size when figures were plotted as separate data points for each aortic segment of a mouse (bar/dot graphs), which has been corrected in the revised manuscript.

References:

- 1 De Moudt S, Leloup A, Van Hove C, De Meyer G and Franssen P (2017) Isometric Stretch Alters Vascular Reactivity of Mouse Aortic Segments. *Front. Physiol.* 8:157. doi: 10.3389/fphys.2017.00157
- 2 Ramírez-Rosas, E., Velázquez, P. N., Verdugo-Díaz, L., Pérez-Armendáriz, E. M., Juárez-Oropeza, M. A., & Paredes-Carbajal, M. C. (2019). Subchronic stress effects on vascular reactivity in C57BL/6 strain mice. *Physiology & behavior*, 204, 283–289. <https://doi.org/10.1016/j.physbeh.2019.03.007>

- 2) Results related to Figure 1: is there any statistically significant difference that should be commented on? Please check throughout the main text, as many statements are not supported by p values, leading to questions about the biological relevance of the results.

Figure 1 is only a representative example, showing the overall protocols used during the study. The idea was to show the all or nothing effects of subjecting an aortic segment to high stretch in the presence and absence of an effective concentration of PE, respectively. The relevance of the results is clear from the further experiments.

3) Any difference between males and females?

We agree that this is a shortcoming of our study as a retrospective analysis on 'sex' is not possible to perform with the current data. Although we admit that there might be differences between both sexes in the absolute values of the responses to the different experimental protocols, our data in males and females never differed in the way the aortic segment responded to the different experimental protocols. Nonetheless, we agree that follow-up studies should certainly include 'sex' as a parameter to further unravel the physiology of aortic de- and re-stiffening after increased pulsatile stretch.

We have added a paragraph in the discussion section to explicitly mention this limitation:

Lines 319-322: *“Sex differences were not systematically looked for in the current study. While there might be differences between both sexes in the magnitude of the responses, our data in males and females never differed in the way the aortic segment responded to the different experimental protocols.”*

Reviewer #2

- 1) In general abstract and introduction parts are not written well and it is hard to understand. The authors should re-write the second half part of the Abstract, and the first page of the Introduction.

We have now made some changes to abstract and the introduction in order to increase the overall readability of these sections.

- 2) In the last sentence of the abstract, the authors should mention the importance of their work in the treatment of which kind of disease.

We have now made changes to the last sentence of the abstract:

Line 49: *“Results of this study may have implications for the therapeutic potential of regular and acute physical activity in the physiology of physical activity and its role in the prevention and treatment of **cardiovascular** disease”*

- 3) Why authors didn't use any control group for their study?

In the scenario they described, a control group treated with an isotonic solution instead of the α_1 adrenoceptor agonist, phenylephrine, would have been a suitable way to evaluate their effect on the parameters and see if there is any difference between the control group and the treated group with PE.

We would like to point out to the reviewer that this study is a complete *ex vivo* study and that mice were not subjected to acute exercise and/or alfa-adrenergic stimulation. In each experiment, each segment of the aorta (multiple per mouse) is its own control, which means that the segments were firstly always conditioned at 80-120 mm Hg in the absence of any chemical stimulation (agonist or antagonist).

Subsequently, the experimental protocol was started, for example: one segment was treated with phenylephrine until a stable contraction was obtained. This segment could then be stimulated with a period of higher stretch (for example 80-170 mm Hg) and the different experimental parameters were followed, calculated and analysed (diameters, compliance and E_p). For clarity, in the absence of PE, no de-stiffening was observed after conditioning at higher pulse pressure.

- 4) In Figure 1 authors mentioned that “ Immediately after the conditioning period of high PP, diameters (figure 1c, d) at 80 and 120 mm Hg were larger than before the conditioning, the diameter change (figure 1e, f) was larger, compliance (figure 1g, h) increased and Petersons elastic modulus (E_p , figure 1i, j) was decreased (which we defined as “de-stiffening”) compared to their isobaric measurement just before the high PP bout. All parameters returned to pre-conditioning values after about 20 to 30 minutes (which we defined as “re-stiffening”)”.

Can the authors elaborate on the duration for which the diameter, compliance, and E_p parameters remained constant at their highest levels after high PP and before re-stiffening? and how was the pattern of returning to preconditioning values?

As shown in figure 1; when returning to normal stretch, changes in diameter, compliance, and E_p were immediate. Further, the pattern of returning to preconditioning values is described in figure 2.

- 5) In figure 6, why did the authors change the PE dose to 100nM?

As we were interested in the **qualitative** more than quantitative stiffening, de-stiffening and re-stiffening values, the dose of PE is not the major determining factor, as long as an effective concentration of PE is used (as shown in figure 2). Indeed, in the experiment of figure 6, a concentration of 100 nM was chosen, as this effectively induces VSMC contraction [1].

References:

- 1) De Moudt S, Leloup A, Van Hove C, De Meyer G and Franssen P (2017) Isometric Stretch Alters Vascular Reactivity of Mouse Aortic Segments. *Front. Physiol.* 8:157. doi: 10.3389/fphys.2017.00157

Reviewer #3

- 1) The western blot on figure 7, it is not clear how the experiment was performed to obtain the fold-change. Also, the "contracted" observation points are always at 1, does this mean that the other represents the relative fold-change from the same tissue as such sample?

The interpretation of the reviewer is correct, as all values represent the fold-change compared to the contracted sample. Similar to the previous experiments, five aortic segments from the same mouse were mounted in the ROTSAC. Four of these segments were contracted with 2 μ M phenylephrine. Afterwards, three of the contracted segments were exposed to increased cyclic stretch (as shown in the figure).

- 2) Figure 6, is there evidence of successful inhibition of Src? Phosphotyrosine inhibitors occasionally fail to inhibit their target, it is recommended always test the proper inhibition of the molecule to avoid making the wrong conclusions.

We agree with the reviewer that we should have tested Src inhibition after PP2 administration (through Western Blotting for example) as this would have provided clear information on the degree of Src inhibition. Here, we used PP2 to inhibit VSMC contraction in a (partially) Src-dependent manner. Indeed, we, and others, have used PP2 in this concentration to successfully inhibit VSMC contraction/adhesion [1-3]. Hence, we didn't perform further analysis in these experiments.

References:

- 1) Gao YZ, Saphirstein RJ, Yamin R, Suki B, Morgan KG. Aging impairs smooth muscle-mediated regulation of aortic stiffness: a defect in shock absorption function? *Am J Physiol Heart Circ Physiol.* 2014 Oct 15;307(8):H1252-61. doi: 10.1152/ajpheart.00392.2014. Epub 2014 Aug 15. PMID: 25128168; PMCID: PMC4200340.
- 2) Neutel CHG, Wesley CD, De Meyer GRY, Martinet W and Guns P-J (2023) The effect of cyclic stretch on aortic viscoelasticity and the putative role of smooth muscle focal adhesion. *Front. Physiol.* 14:1218924. doi: 10.3389/fphys.2023.1218924
- 3) Sun Z, Martinez-Lemus LA, Hill MA, Meininger GA. Extracellular matrix-specific focal adhesions in vascular smooth muscle produce mechanically active adhesion sites. *Am J Physiol Cell Physiol.* 2008 Jul;295(1):C268-78. doi: 10.1152/ajpcell.00516.2007. Epub 2008 May 21. PMID: 18495809; PMCID: PMC2493553.

- 3) Figure 4 A, the letter A needs a better description.

We have made changes in the manuscript to address this comment. In this experiment we kept the pulse pressure constant at 40 mm Hg, but changed the diastolic and systolic pressure. The figure legend now read as follows:

Line 758-761: *“Figure 4. The role of mean pressure in aortic tissue de-stiffening. Relative E_p in the presence of 300 μ M L-NAME and 2 μ M PE as a function of time at 80-120 mm Hg after conditioning the segments for 5 minutes at different pressures. **Experimental protocol showing that the pulse pressure was kept constant (at 40 mm Hg) whilst the mean pressure was increased: 80-120, 100-140, 120-160 and 140-180 mm Hg. Data were compared with the higher pulse pressure of 90 mm Hg (80-170 mm Hg) (A). E_p was expressed in % with E_p before the conditioning period as 100% (B). Curves were fitted with a mono-exponential function revealing amplitude of de-stiffening (at 50 sec in the graph, when segments were clamped between 80 and 120 mm Hg) (C), amount of re-stiffening (D) and time constants of re-stiffening (E). The box on plot B represents the conditioning period for 5 minutes at 80-120 mm Hg, 100-140 mm Hg, 120-160 mm Hg and 140-180 mm Hg. *, **, ***: $p < 0.05, 0.01, 0.001$ versus 80-170 mm Hg ($n=5$). E_p = Peterson's Modulus of Elasticity”***

REVIEWERS' COMMENTS:

Reviewer #1 (Remarks to the Author):

My concerns have been addressed, thank you.

Reviewer #2 (Remarks to the Author):

Thanks for the response by the authors. They made acceptable changes to the abstract, and the answers to questions were acceptable. I have no further questions.

Reviewer #3 (Remarks to the Author):

The work has great value for the field. Taking into account their limitation, the experimental design was adequate to address the scientific questions. After a detailed review, I detected a few details that require clarification and revision, which were addressed by the authors at a big extent.

Given the stated above, I agree in publishing the manuscript if authors and editors agree as well.